# The RNA Methyltransferase NSUN2 and Its Potential Roles in Cancer

**DOI:** 10.3390/cells9081758

**Published:** 2020-07-22

**Authors:** Anitha Chellamuthu, Steven G. Gray

**Affiliations:** 1Department of Clinical Medicine, Trinity College Dublin, Dublin D08 W9RT, Ireland; chellama@tcd.ie; 2Thoracic Oncology Research Group, St. James’s Hospital, Dublin D08 RX0X, Ireland

**Keywords:** 5-methylcytosine, miRNA, mRNA, RNA modification, epi-transcriptome, cancer

## Abstract

5-methylcytosine is often associated as an epigenetic modifier in DNA. However, it is also found increasingly in a plethora of RNA species, predominantly transfer RNAs, but increasingly found in cytoplasmic and mitochondrial ribosomal RNAs, enhancer RNAs, and a number of long noncoding RNAs. Moreover, this modification can also be found in messenger RNAs and has led to an increasing appreciation that RNA methylation can functionally regulate gene expression and cellular activities. In mammalian cells, the addition of m5C to RNA cytosines is carried out by enzymes of the NOL1/NOP2/SUN domain (NSUN) family as well as the DNA methyltransferase homologue DNMT2. In this regard, NSUN2 is a critical RNA methyltransferase for adding m5C to mRNA. In this review, using non-small cell lung cancer and other cancers as primary examples, we discuss the recent developments in the known functions of this RNA methyltransferase and its potential critical role in cancer.

## 1. Introduction

The first intimation that RNA could be modified came from studies in the 1960s which demonstrated that pseudouridine (5-ribosyluracil, or ψ) could be incorporated into transfer RNA (tRNA) and ribosomal RNA (rRNA) [1,2]. Since then, a large number of posttranscriptional modifications of RNA have been described, which currently stands at more than 170 different modifications [3,4]. This is in contrast to the known modifications found in DNA, which number approximately 20 [5,6].

The most commonly found RNA modification in mRNA is called N6-methyladenosine (m6A) and was first described in 1974 by Desrosiers et al. [7]. Technological issues hampered the early investigation of this RNA modification [8]. However, in the last decade, methodologies have emerged, which have mapped the presence of this modification in a large number of mRNA sites [9,10]. In addition, the discovery of proteins which can act as specific “readers”, “writers”, and “erasers” for m6A and the discovery that a histone H3 (H3K36me3) chromatin mark guides m6A deposition in the coding sequence (CDS) and 3′-untranslated regions (UTRs) have led to the term “epitranscriptomics” to describe the role of RNA modifications in cellular processes [11]. 

5-methylcytosine (m5C) has frequently been associated with tRNA [2] but was first identified in messenger RNA (mRNA) in the 1970s [12,13], and subsequently, more recent studies have identified that this modification is found most often in 5’ and 3’-UTRs of mRNAs, with a pronounced peak in the vicinity of the translational start codon [2,14,15]. 

Early methodologies to identify m5C utilized liquid chromatography mass spectrometry (LC-MS) but can be considered currently unsuitable for analysis of m5C for technical reasons [2]. Since then, new technological developments have been developed to identify m5C in RNA. 

One of the most established is RNA-bisulfite sequencing. Bisulfite sequencing is currently the most accurate and reliable method to detect m5C marks at nucleotide resolution in both DNA and RNA and basically involves the chemical conversion of unmethylated cytosine to uracil. Methylated cytosines are protected from this conversion, allowing users to determine m5C methylation at the singe nucleotide level by sequencing [16]. With the advent of next-generation sequencing (NGS) coupled with bisulfite conversion, large-scale analysis of m5C in DNA and RNA can be assessed, often called RNA-BS-Seq [2]. Early difficulties in measuring RNA m5C involved issues with (a) incomplete conversion artifacts and (b) misamplification in the RNA, both of which could obscure actual methylation signals, and (c) computation methodologies at the time, which were focused on the assessment and identification of m5C at DNA CpG residues [2,16]. However, these technological issues have now mostly been resolved [2], and computational pipelines have now been developed for specific analysis of m5C in RNA [16].

Other methodologies based on NGS have also been developed. One, methylated RNA-immunoprecipitation sequencing (meRIP-Seq) was initially developed to study changes in m6A [10]. This technology basically involves the use of an antibody specific to the modification being examined and involves immunoprecipitation followed by sequencing. meRIP-Seq has been used successfully to study m5C in *Arabidopsis* [17] and in bacteria, archaea, and yeast [2,18] and, most recently, has been utilized in the study of m5C associated with mRNA in matched normal-tumor hepatocellular carcinoma (HCC) samples [19]. Two other technologies have been developed which allow research to examine for changes in m5C (or other methylation marks) that involve mechanism-based enrichment procedures specific to individual RNA methyltransferases (RNMT). The first, known as azacytidine immunoprecipitation (aza-IP), allows for the study of m5C. In essence, the technique is based on the fact that all known mammalian m5C-RNMTs form a reversible covalent intermediate with their cytosine substrate. However, by using a cytosine analog 5-azacytidine (5-aza-C), the RNMTs form a covalent linkage at sites of cytosine incorporation but become trapped and can be detected following immunoprecipitation (IP) with an antibody specific to the RNMT and high-throughput sequencing to identify direct RNA substrates of individual m5C RNMTs [20], although concerns exist regarding sensitivity and quantitativeness [2]. Methylation-individual nucleotide resolution crosslinking immunoprecipitation (miCLIP) is another technology used to study m5C. This technology is based off crosslinking immunoprecipitation (CLIP), a stringent technique devised to identify RNA–protein interactions and uses UV crosslinking to induce a covalent bond between protein and RNA [21]. When combined with immunoprecipitation and NGS, this allows for genome-wide analysis of crosslink sites at individual nucleotide resolution or iCLIP [22]. By using immunoprecipitation specific to m5C RNMTs (in this instance, the RNMT NSUN2), Frye and colleagues were able to successfully interrogate NSun2-mediated cytosine-5 methylation in RNA [23]. Using these techniques, researchers have identified multiple RNA species that contain m5C [14,15,24,25,26].

## 2. The Family of m5C RNA Methyltransferases

At present, the known mammalian m5C RNA methyltransferases (RNMTs) belong to either the DNA Methyltransferase family (specifically TRDMT1/DNMT2) [27] or belong to the NSUN (NOL1/NOP2/sun domain) family (NSUN1–NSUN7) [28] (Figure 1).

Both of these families utilize S-adenosyl-L-methioinine (SAM) as a donor for the transfer of methyl groups to RNA, and the mechanisms by which these proteins catalyze this transfer have been recently well reviewed [2,32]. Whilst the majority of RNA methyltransferases have been shown to play roles in methylating rRNA (NSUN1/NSUN4/NSUN5) [32], tRNA (NSUN2/NSUN3/NSUN5/NSUN6 and DNMT2) [32], mitochondrial tRNAs (NSUN2/NSUN3) [32,33] and enhancer RNAs (eRNAs) (NSUN7) [32] and while roles for RNMTs in methylating mRNAs, miRNAs and long noncoding RNAs (lncRNAs) have been predominately ascribed to NSUN2 [26,34,35,36], a recent report has suggested that TRDMT1 may also methylate mRNAs [37]. As the purpose of this review is to discuss NSUN2, the reader is directed to the following reviews for additional reading of the functional roles of the other known RNMTs [2,32].

### 2.1. Known Cellular Roles for NSUN2-Directed RNA Methylation

A large body of evidence has now shown that NSUN2-mediated RNA methylation plays many cellular functions and are summarized in Table 1.

#### 2.1.1. Direct Functions of NSUN2

NSUN2 is a predominantly nucleolar protein and plays important roles in tissue homeostasis, spindle stability, and early embryogenesis [38]. During mitosis at the interphase stage, NSUN2 moves as an RNA–protein complex with nucleolar and spindle-associated protein (NuSAP) from the nucleoli to the mitotic spindle [39].

#### 2.1.2. Indirect Functions of NSUN2

Various indirect functions for NSUN2 in cellular processes have been described and include cellular differentiation, testis differentiation, and early embryogenesis [40,41,42,43,44,45]. In particular, NSUN2 has been shown to increase protein production by diverse mechanisms including promoting mRNA stability [46,47], affecting miRNA maturation [48], mRNA nuclear exporting [36], altering gene and lncRNA expression [34,49], and enhancing protein synthesis and translation [26,50,51,52]. NSUN2 also plays roles in cellular proliferation where it affects the expression and translation of key cell cycle regulators [46,53,54,55,56]. Alternatively, it can also act as a sensor under various stresses, inducing situations to regulate cellular senescence [50,56,57]. Taken together, the full functional roles of m5C methylation of RNA as catalyzed by NSUN2 have yet to be fully delineated, but significant evidence suggests that such methylation events have important cellular roles which may be important in disease pathogenesis. 

## 3. Known Roles for NSUN2 in Disease

A significant body of evidence now links m6A RNA methyltransferases with both memory and neural development [60,61,62,63,64,65,66,67,68,69,70] and in pathological settings such as Alzheimer’s disease [71]. Early indications that m5C RNA methyltransferases could also play roles in neural settings came from knockout models. Loss of NSUN2 in Drosophila was associated with severe short-term memory deficits [72], while deletion of NSUN2 in mice led to impaired neural development with inhibition of neural cell migration and impaired differentiation of neuronal stem cells, resulting in an accumulation of intermediate progenitors within the developing cerebral cortex and a loss of upper layer neurons [42]. Moreover, if DNMT2 and SUN2 are simultaneously deleted, they demonstrate a synthetic lethal interaction, which included a reduced thickness and organization of the cerebral cortex [52]. One of the features of loss of m5C RNMTs is elevated tRNA degradation [41,73], leading to an accumulation of 5′ tRNA-derived small RNA fragments. This results in reduced protein translation rates and activates stress pathways (such as oxidative stress), leading to reduced cell size and increased apoptosis of cortical, hippocampal, and striatal neurons [41,52]. Recently, NSUN2 has been shown to methylate mitochondrial tRNAs [33,74], and given the major role of mitochondria in neurological disorders involving neurodegeneration [75,76], it is perhaps conceivable that m5C RNA methylation deficiencies in both the nuclear and mitochondrial settings may play roles in neural disease. In this regard, mutations of NSUN2 have been associated with autosomal recessive disorders associated with intellectual disability including Dubowitz syndrome (DS) and autism spectrum disorders (ASD) [72,77,78,79,80,81]. Most recently Kim et al. report that NSUN2 is dysregulated in the brains of patients with Alzheimer’s disease (AD) [82]. In primary neuronal cultures, oligomeric amyloid beta (Aβ) induced both the dysregulation of NSUN2 and altered tau proteostasis while overexpression of NSUN2 could rescue tau-induced toxicity [82].

Aside from neurological disorders, some emerging evidence now links the activities of both m6A and m5C RNMTs in the vascular setting. Various m6A RNMTs have now been shown to have roles in endothelial cells, vascular calcification, and abdominal aortic aneurysm [83,84,85,86,87]. Regarding m5C RNMTs, DNMT2 has been shown to be upregulated by curcumin in vascular smooth muscle cells and this was associated with RNA stabilization [88]. NSUN2 has been shown to methylate SHC Transforming Protein (SHC) mRNA, resulting in the enhanced translation of SHC isoforms under conditions of stress (oxidative or high glucose) and causing an accelerated senescence of human vascular endothelial cells (HUVECs) [50]. Moreover, methylation of Intercellular Adhesion Molecule 1 (ICAM-1) mRNA by NSUN2 promotes enhanced translation of this adhesion molecule and results in increased adhesion of leukocytes to endothelial cells [89]. Furthermore, loss of NSUN in a rat knockout model was associated with impairment of allograft arteriosclerosis development [89].

## 4. Known Roles for NSUN2 in Cancer

A significant body of work has now shown that RNMTs are aberrantly expressed and play important roles in the development and pathogenesis of cancer [90,91]. In this regard, an early study of NSUN2 found that it was linked with the oncogene MYC protooncogene, bHLH Treanscription Factor (MYC), whereby NSUN2 is upregulated upon Myc activation, and moreover, NSUN2 is highly expressed in various tumors [92,93]. A subset analysis of The Cancer Genome Atlas (TCGA) datasets confirms that the expression of NSUN2 is frequently overexpressed at the mRNA level in various cancers (Figure 2), as are various other members of the m5C RNMTs (Appendix A). Over the following sections, we describe the emerging data regarding the role of NSUN2 in cancer.

### 4.1. NSUN2 and Breast Cancer

Several studies have now shown that NSUN2 is overexpressed in breast cancer. In their original report on NSUN2, Frye and Watt observed that NSUN2 was overexpressed in 7/7 human breast carcinomas [92]. Using array-comparative genomic hybridization (a-CGH), Frye and colleagues subsequently demonstrated that, in a panel of breast cancer cell lines, 18 out of 50 cell lines analyzed had a copy number gain of the genomic region containing NSUN2 (5p15.31-33), and of these, 16 (16/18) were found to have significantly increased expression of NSUN2 [95]. The region 5p15 is associated with a strong risk for the development of breast cancer [96]. In primary tumors, 19% of tumors (23 of 119) showed a gain of NSUN2 while 7% (8 of 119) had a loss of NSUN2 and there was no significant correlation between the level of NSUN2 and tumor subtype (luminal A or B, basal, Her2, or normal-like) [95]. Immunohistochemical analysis of tissue microarrays found that protein expression of NSUN2 was upregulated in 34% of analyzed breast tumors (*n* = 89 tumors), but that this upregulation showed no correlation with the expression of Ki67, Estrogen Receptor (ER) status, or MYC amplification [95]. The upregulation of NSUN2 at the protein level in breast cancer was subsequently confirmed by Tatsuka and colleagues [93]. DNA hypomethylation has also been found to be associated with increased NSUN2 expression in breast cancer, and NSUN2 overexpression promoted cell proliferation, migration, and invasion while NSUN2 knockdown inhibited these processes in vitro and in vivo [97]. A more detailed pan-cancer analysis has determined that expression of several RNMTs, including NSUN2, had positive correlations between DNA copy number and mRNA expression and was associated with higher grade aggressive subtypes and poor prognosis in breast cancer patients [98]. While the initial studies of NSUN2 in breast cancer found no association with ER status or Ki67 [95], in contrast, the study by Yi et al. found significant correlations with estrogen receptor, Ki-67, and the progesterone receptor [97]. These studies clearly demonstrate the important role of NSUN2 in breast cancer, but additional work will be required to clarify the discrepancies found between various clinical parameters.

### 4.2. NSUN2 and Colorectal Cancer

In addition to their original description of NSUN2 as an RNMT, Frye and Watt also observed high expression of NSUN2 in 3/4 colon cancers [92], an observation subsequently confirmed by other researchers [93]. In their study of Proteinase activated-receptor 2 (PAR2)-mediated cancer cell migration, Wang and colleagues identified NSUN2 as a critical protein in this process. In this study, PAR2-signaling suppressed miR-125b via NSUN2. NSUN2 functions to regulate this miRNA by methylating the precursor pri-miR-125b2. This methylation inhibits the processing of pri-miR-125b2 into pre-miR-125b2, decreases the cleavage of pre-miR-125b2 into miR-125, and attenuates the recruitment of RISC by miR-125, thereby repressing the function of miR-125b in silencing gene expression [48]. This in turn resulted in the elevation of its target Grb2-associated-binding protein 2 (GAB2), which was identified as the protein mediating the pro-migratory effects [59].

Circular RNAs (circRNAs) are a class of noncoding RNA (ncRNA) that, unlike linear RNAs, form covalently closed continuous loops and act as gene regulators in mammals [99]. Intriguingly a circRNA derived from the NSUN2 coding sequence called circNSUN2 (hsa_circ_0007380) has recently been identified as being frequently upregulated in patients with colorectal carcinoma (CRC) and predicts poorer patient survival [100]. When overexpressed, circSUN2 was shown to promote liver metastases in patient-derived xenograft (PDX) models. Moreover, m6A methylation of this circRNA was shown to be essential to this process. Once methylated, circNSUN2 is exported by YTH domain-containing protein 1 (YTHDC1) from the nucleus to the cytoplasm. Once there, circNSUN2 interacts with Insulin-Like Growth Factor 2 mRNA-Binding Protein 2 (IGF2BP2), an RNA-binding protein (RBP) [100]. A search by the authors for target RNAs which interact with IGF2BP2 identified high mobility group AT-hook 2 (HMGA2) mRNA as a likely candidate [100]. Given the known roles of HMGA2 in inducing epithelial-to-mesenchymal transition (EMT) and in contributing to colon cancer progression [101,102,103], the authors subsequently went on to show that the circSUN2/IGF2BP2 complex bound to HMGA2 mRNA in an RNA–protein ternary complex that stabilizes HMGA2 mRNA and results in the promotion of EMT in CRC cells [100].

### 4.3. NSUN2 and Lung Cancer

The current available published evidence for altered expression of m5C RNMTs in lung cancer is sparse. Due to its original identification as a marker of proliferation [91,104], high expression of NSUN1 (NOP2/p120) has been identified as a prognostic biomarker in non-small cell lung cancer (NSCLC) [105,106,107]. Likewise, TRDMT1 (DNMT2) was reported to have a higher activation in small cell lung cancer (SCLC), neuroblastoma, and medulloblastoma compared to all other cancers [108]. Loss of the region containing NSUN3 has been reported as a common loss in lung adenocarcinomas of never-smokers at a frequency of 15% [109], and interestingly, this was the only coding gene in the region lost [109]. Mutation of NSUN3 has been shown to result in reduced mitochondrial protein translation and respiration [2], while some evidence has shown a role for NSUN3 in nuclear transcriptional regulation in leukemic cells [110].

During early embryogenesis, NSUN2 is constitutively expressed [43], but it remains unsure what its role, if any, is in the developing lung. In NSUN2 knockout mice, the mice develop normally but are phenotypically 30% smaller than normal and show defects in skin and testis development [44,45].

Most recently, a pan-cancer analysis of RNMTs has shown that alterations to NSUN2 are common in lung cancer [98], although this was not followed up in more detail. In this regard, cBioPortal analysis [111,112] of the TCGA lung adenocarcinoma (LUAD) and lung squamous cell carcinoma (LUSC) datasets indicates that alterations to NSUN2 were found in 37% and 44% of samples, respectively (Figure 3).

cBioPortal analysis [111,112] was used to examine for alterations in the TCGA PanCancer Atlas datasets [113] comprising LUAD (*n* = 566) and LUASC (*n* = 487) samples. Mutations, putative copy number alterations, and mRNA expression were selected for analysis using default settings. The results are presented as OncoPrints, a compact means of visualizing distinct genomic alterations, including somatic mutations, copy number alterations, and mRNA expression changes across a set of cases. Given the large number of alterations observed in the cBioPortal analysis, we subsequently conducted additional analyses for NSUN2 using Lung Cancer Explorer (LCE) [114]. 

For tumor-normal gene expression difference, Lung Cancer Explorer generates searchable tables for LUAD and LUSC that contain the summary statistics from meta-analysis of standardized mean difference (tumor–normal) using Hedges’ G as an effect size metric. Studies included for meta-analysis must have at least 10 samples in each group, and meta-analysis was only performed for genes with data available from at least three qualifying studies [114]. For survival association with gene expression, Lung Cancer Explorer generates searchable tables that contain the summary statistics from meta-analysis of survival and gene expression association based on the univariate Cox Proportional–Hazards Model. The gene expression in each study was normalized to zero mean and unit variance for each gene before the model was fitted. Only studies with at least 10 samples that have survival data were included, and meta-analysis was only performed for genes with data available from at least three qualifying studies [114]. Systematic analysis of NSUN2 expression for associations with (a) tumor-normal gene expression differences and (b) overall survival (OS) were conducted, and significant associations for altered NSUN2 expression in tumors were observed for both LUAD and LUSC while a significant survival association was observed for LUAD (Table 2).

Meta-analysis of all available datasets on LCE for gene expression of tumor vs. normal and survival are also presented graphically in Figure 4. The meta-analysis provided by LCE effectively combines the statistical strength from multiple data sets and allows greater precision than using any of the single studies. Using LCE to examine all NSCLC datasets in their database, forest plots were generated to summarize tumor-normal standardized mean differences for tumor vs. normal meta-analysis (Figure 4A) and hazard ratios for survival meta-analysis, as shown in Figure 4B.

Given the significant alterations observed in the meta-analysis for tumor versus normal gene expression changes, we reanalyzed the TCGA datasets using UALCAN [94]. In agreement with the previous analyses, significant alterations were observed in both datasets for normal vs. tumor and when stratified by tumor stage, race, gender, age, smoking habit, nodal metastasis, and p53 mutation status, and the results are presented in Appendix A. In addition, protein analysis confirmed significant protein overexpression in lung adenocarcinomas using the Clinical Proteomic Tumor Analysis Consortium (CPTAC) Confirmatory/Discovery dataset (Figure 5).

As a strong association was observed for NSUN2 with overall survival in LUAD, we also examined the prognostic value of NSUN2 mRNA using KM-Plot [115] (Figure 6A). Overall, a high expression of NSUN2 was associated with a better OS. When patients were stratified based on histology, it was shown that a high expression of NSUN was associated with significantly better OS in just the adenocarcinoma subtype (Figure 6B) and not in the squamous cell carcinoma subtype (Figure 6C), in agreement with the lung cancer explorer meta-analysis (Table 2).

The data presented here clearly demonstrate that NSUN2 is significantly overexpressed/altered in NSCLC, and while evidence for a functional role for this protein in lung cancer is currently lacking, it would suggest that this protein may play important roles in lung cancer pathogenesis and suggests that its expression may also have utility as a prognostic marker.

### 4.4. NSUN2 and Other Cancers

The roles for m5C RNMTs have now been identified in several other cancers. In one early study, elevated protein expression of NSUN2 was confirmed in various cancers including esophageal, stomach, liver, pancreas, uterine cervix, prostate, kidney, bladder, thyroid, and breast cancer [93]. In skin cancer, loss of NSUN2 expression was correlated with increased malignancy when protein expression levels in normal skin and cutaneous cancers were compared across increasing tumor/node/metastasis (TNM) stages [116] and was associated with an increase of tumor-initiating cells [116]. In ovarian cancer, an NSUN2/IGF2 signature has been identified, which has prognostic survival value, as patients with high NSUN2 expression and low IGF-II expression had significantly superior overall and disease progression-free survival [117]. A recent meta-analysis study in Head and Neck Squamous Carcinoma (HNSCC) identified that NSUN2 had approximately 2-fold upregulation in tumors, which was associated with 22 months shorter overall survival and a higher mortality risk [118]. Moreover, high NSUN2 expression has been associated with T-cell activation in HNSCC, and there was a positive association between T-cell activation score and mortality in patients with low NSUN2 expression, which suggests that NSUN2 expression in HNSCC could be used as a marker to stratify patients for immune-checkpoint blockade [119].

In bladder cancer, Y-Box Binding Protein 1 (YBX1) has been identified as a an m5C “reader” and was shown to promote carcinogenesis by stabilizing Hepatoma-Derived Growth Factor (HDGF) through an interaction with Elav-Like RNA-Binding Protein 1 (ELAVL1). Importantly, high co-expression of NUSN2, YBX1, and HDGF was found to predict the poorest survival in urothelial carcinoma of the bladder (UCB) [47]. High expression of NSUN2 was found to occur in gallbladder carcinoma (GBC), and silencing of NSUN2 repressed GBC cell proliferation and tumorigenesis both in vitro and in vivo via an interaction with Ribosomal Protein L6 (RPL6) [120]. Finally, a very recent study has identified a role for NSUN2 in gastric cancer promoting cancer cell proliferation both in vitro and in vivo via repression of the cyclin dependent kinase p57^Kip2^ in an m5C-dependent manner [54].

### 4.5. Therapeutic Considerations for Targeting NSUN2 Associated Cancer Pathways

Sphingosine kinases have been shown to be potential targets for therapy in breast cancer [121,122]. In this regard, knockdown of Sphingosine kinase 1 (SK1) was associated with downregulation of NSUN2 in breast and prostate cancer cells [123], suggesting that pharmacological inhibitors of SK1 could potentially benefit cancers with overexpressed NSUN2 [124]. At the same time, directly knocking down NSUN2 in HeLa cells has been shown to potentiate the sensitivity of the cells to 5-fluorouracil (5-FU) but not that of cisplatin or paclitaxel [125]. In esophageal cancer cells, NSUN2 was found to associate with and methylate an lncRNA, NMR (also known as LINC01672 or lnc-CAMTA1-1:2/CAMTA1-IT1) [35], for which expression is associated with resistance to cisplatin or paclitaxel [35]. 

Vaults are ribonucleoprotein complexes widely associated with the induction of chemoresistance and long-term chemotherapy failure [126] and for which expression plays a role in preventing apoptosis and regulating autophagy [127,128,129,130]. Vault complexes are composed primarily of three proteins: a 100-kDa major vault protein (MVP), a 290-kDa telomerase-associated protein 1 (TEP1), and a 193-kDa vault poly (ADP-ribose) polymerase (VPARP) complexed with noncoding vault RNA (vtRNAs) [126]. vtRNAs have been shown to have the ability to recognize chemotherapeutic compounds, such as mitoxantrone, and it has been suggested that the binding of vtRNAs to these chemotherapeutic compounds prevents the drugs from reaching their target sites and consequently in cancer chemoresistance [131,132]. Given that NSUN2 has been shown to exclusively methylate vtRNAs (and in particular vtRNA1.1) [40] and that loss of m5C in vault RNAs causes aberrant processing of vtRNAs [23], targeting NSUN2 may represent a potentially new way to overcome resistance.

Molecularly targeting m5C RNMTs using drugs previously developed to target DNMTs is a potential possibility as demonstrated from using Aza-IP [24] and preliminary in vitro data in leukemic cells, where novel RNA:m5C/RCMT-mediated chromatin structures form that modulate 5-AZA response/resistance [110]. Alternatively, it may be possible to target NSUN2-mediated events such as that shown for the lncRNA NMR in esophageal cancer, whereby NSUN2-methylated NMR interacts with the chromatin remodeling factor Bromodomain PHD Finger Transcription Factor (BPTF) [35,118] to regulate resistance to cisplatin. In this regard, a specific inhibitor for BPTF has been identified [133], and shown to be effective in BTPF expressing lung cancer. BPTF is overexpressed in NSCLC and is associated with poor prognosis [134,135], and as cisplatin is a mainstay of NSCLC therapy, it will be important moving forward to determine if expression of either NSUN2 or BTPF is associated with resistance to cisplatin. Furthermore, additional studies will be required to assess if C620-0696 could sensitize patients to cisplatin therapy. In addition to BPTF, YBX1 has also been shown to interact with and to stabilize mRNAs methylated by NSUN2 [47]. In this regard, the YBX1 protein has been shown to be overexpressed in cancer cells resistant to cisplatin therapy [136], and in a recent study, YBX1 has been shown to mediate sensitivity to cisplatin in a xenograft model of NSCLC [137]. However, in order for YBX1 to translocate to the nucleus to bind and stabilize mRNAs such as HDGF [47], it must first translocate to the nucleus. This is normally achieved via phosphorylation of serine 102 by various kinases such as AKT Serine/Threonine Kinase AKT [138]. Recently, treatments with agents such as the multikinase inhibitor TAS0612 or with everolimus, an mammalian target of the rapamycin complex 1 (mTORC1) inhibitor, to prevent YBX1 phosphorylation have been shown to overcome YBX1-mediated drug resistance [139].

Finally, the identification of NSUN2 expression as a potential biomarker for stratifying patients for immune-checkpoint inhibitors at least in HNSCC [119] is a finding that should be assessed in other tumors moving forwards.

Several companies have emerged to target RNA Epigenetics. At the present moment in time, many of these are focused on targeting m6A RNMTs [140], but given the rapid advances in the field of RNMTs, targets for m5C RNMTs will presumably also be on these companies research and development pipelines.

## 5. Conclusions

Whilst still a relatively new field, the recent discoveries of the role of the epitranscriptome in cancer clearly links RNMTs as a candidate therapeutic target moving forwards. While m6A RNMTs are currently the focus of several targeting strategies in the pharmaceutical setting, emerging evidence now begins to link m5C RNMTs to cancer.

Targeting the proteins that add m5C modifications to DNA (particularly DNA methyltransferases) has been an area of active research in cancer. Agents such as 5-azacytidine (vidaza) or 5-aza-2-deoxycytidine (decitabine) originally developed to target these proteins have been approved by the U.S. Food & Drug Administration (FDA) [141,142] for the treatment of myelodysplastic syndromes (MDS), but success in translating these drugs into the treatment of solid tumors has been limited. Recent studies have suggested that, rather than using cytotoxic dosage regimens, “epigenetic priming” using clinically relevant nontoxic doses may have better efficacy [143,144]. Over the previous sections, we have elaborated on the known molecular functions attributable to the m5C RNMTs, with particular attention to NSUN2. It has become increasingly clear that this RNMT plays important roles in cancer development. Given that alterations involving increased expression of NSUN2 are common in breast cancer [93,95,98], colon cancer [92,93], and lung cancer as suggested in the suggestions above, molecularly targeting this RNMT would appear to an attractive approach moving forward. While studies on NSUN2 have used azacytidine in the past [24,110], due to the previously poor responses to this drug observed in the past, newer approaches may be necessary to elicit effective responses in patients. For example, it may be possible to use agents targeting NSUN2 to increase the number of patients that may be suitable for checkpoint inhibitors. Other inhibitors may be suitable candidates to be brought forward into clinical trials as NSUN2 belongs to the Rossmann fold family of methyltransferases which includes the histone lysine methyltransferase DOT1L. In this regard, an inhibitor of DOT1L (pinometostat) has recently entered clinical trials in leukemia [91,145], and it may be possible to see if this agent also inhibits NSUN2. Large consortia such as the Structural Genomics Consortium (SGC) may identify novel agents capable of targeting this RNMT [146], or indeed, companies targeting RNA epigenetics may develop candidate leads to move forward in the clinic [140].

## Figures and Tables

**Figure 1 cells-09-01758-f001:**
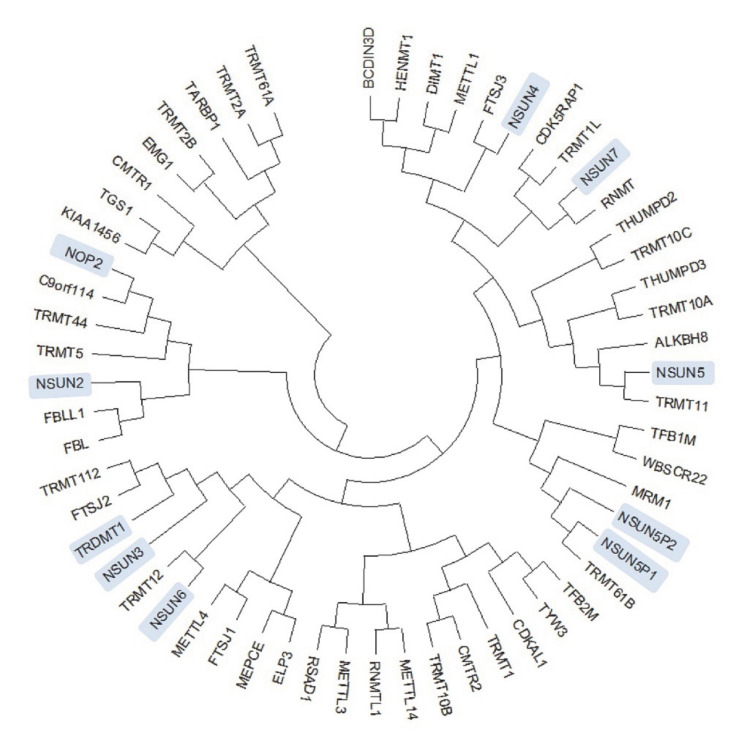
Phylogenetic analysis of RNA methyltransferases (RNMTs): Analysis was conducted using MEGA-X [29]. The sequences of 58 RNMTs were obtained from the ChromoHub database [30] and aligned, and an evolutionary phylogenetic tree was inferred by using the maximum likelihood method and Jones, Taylor & Thornton (JTT) matrix-based model [31], using default parameters. Members of the m5C RNMTs are highlighted in blue.

**Figure 2 cells-09-01758-f002:**
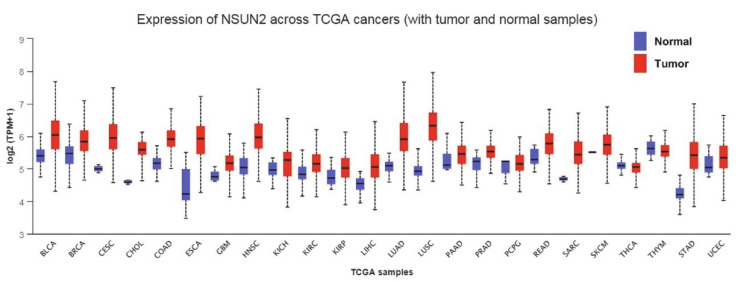
Pan-cancer expression of NSUN2 (normal vs. tumor) as assessed using UALCAN [94]. Altered expression of NSUN2 can be seen across many cancers.

**Figure 3 cells-09-01758-f003:**
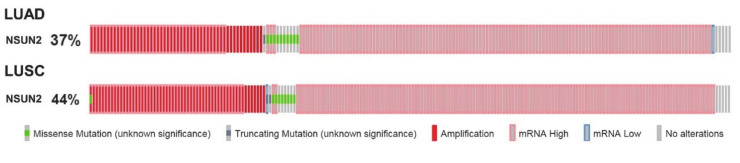
cBioportal analysis of The Cancer Genome Atlas (TCGA) lung adenocarcinoma (LUAD) and lung squamous cell carcinoma (LUSC) datasets.

**Figure 4 cells-09-01758-f004:**
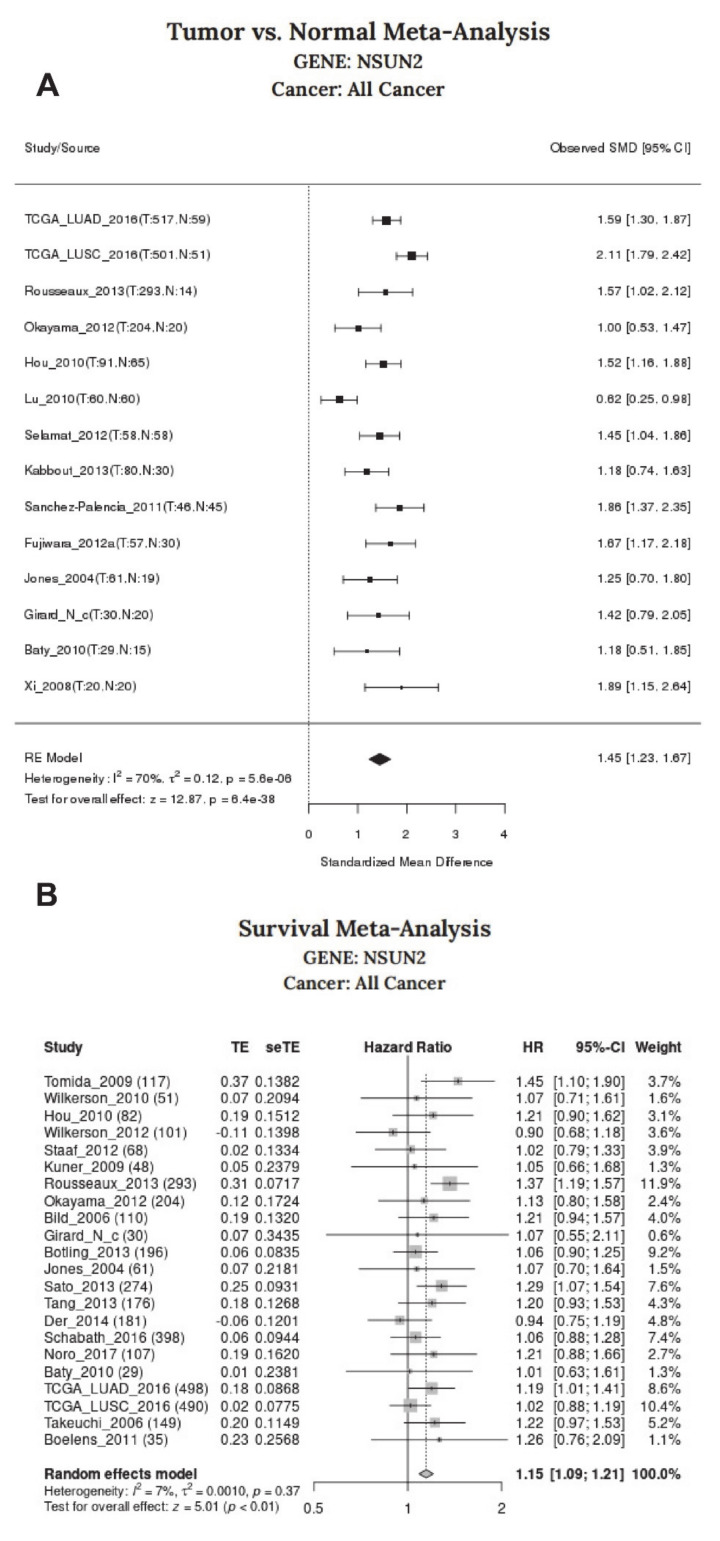
Meta-analysis of NSUN2 for changes in gene expression and survival: All available datasets on Lung Cancer Explorer [114] were assessed by meta-analysis for (**A**) associations between gene expression of tumor vs. normal and (**B**) survival using the default settings in the online server.

**Figure 5 cells-09-01758-f005:**
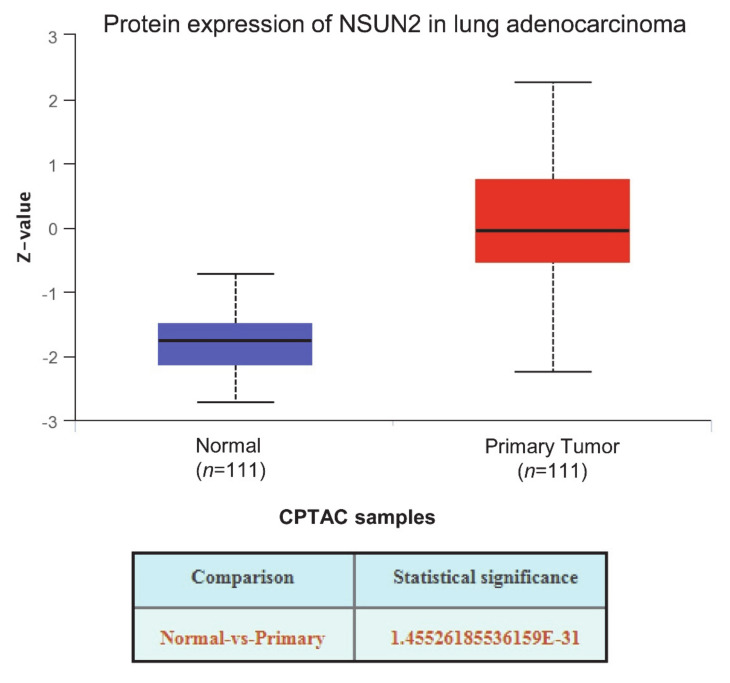
NSUN2 protein is overexpressed in LUAD samples: The expression of the NSUN2 protein was examined in the Clinical Proteomic Tumor Analysis Consortium (CPTAC) Confirmatory/Discovery dataset for LUAD using UALCAN [94]. Significant overexpression of NSUN2 protein was observed in tumors compared to normal tissues.

**Figure 6 cells-09-01758-f006:**
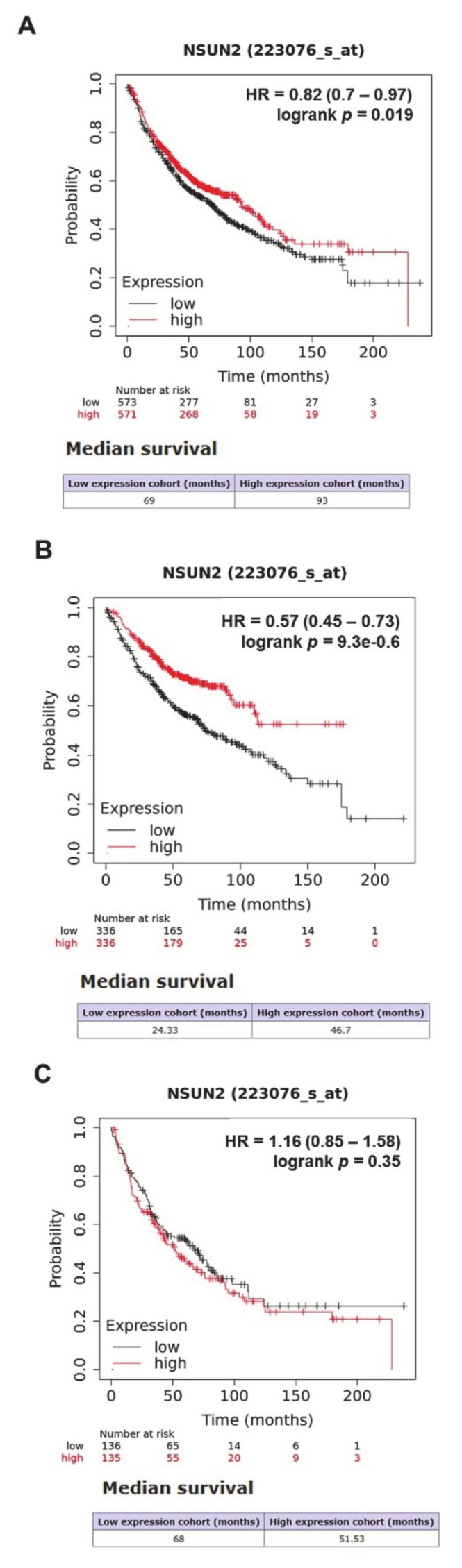
NSUN2 has a potential prognostic value in lung cancer. The potential prognostic value of NSUN2 mRNA expression was evaluated using KM-Plot [115]. Patients were split using median expression, and survival was assessed using the default parameters. Analyses were conducted based on (**A**) all histologies, (**B**) LUAD, and (**C**) LUSC.

**Table 1 cells-09-01758-t001:** Functional roles for NOL1/NOP2/SUN domain (NSUN2)-mediated RNA methylation.

Role of Nsun2	Biological Function	Reference
Mitosis	Mitotic spindle stability	[39,58]
Cellular Proliferation	Affects expression of p16^INK4^/CDKN2A, p57^kip2^, and translation of CDK1 and CDKN1B (p27^kip1^)	[46,53,54,55,56]
Cellular Senescence	NSUN2-mediated m5C alters translation of CDKN1B (p27^kip1^) and CDK1 resulting in senescence.	[56]
	Methylation of SHC mRNA by NSUN2 results in activated p38MAPK and cellular senescence under high oxidative stress- or high-glucose conditions.	
	Under oxidative stress conditions, NSUN2-directed m5C of p21 mRNA enhances its translation, leading to elevated expression of p21 and cellular senescence.	
Cellular Migration	NSUN2-dependent mediated regulation of pri-miR-125b processing affects Proteinase-activated Receptor 2 (PAR2)-mediated cell migration.	[48,59]
Cellular Differentiation	NSUN2-mediated m5C methylation of small-vault RNAs (svRNAs) affects epidermal differentiation.	[40]
	NSUN2 is expressed in early neuroepithelial progenitors of the developing human brain, and its expression is gradually reduced during differentiation of human neuroepithelial stem (NES) cells.	
	NSUN2 expression is developmentally regulated during embryogenesis with a possible role in body axis extension.	
	NSUN2 is essential for germ cell differentiation in mouse testis.	
	NSUN2 -/- mice show neurodevelopmental deficiencies.	
mRNA Nuclear Export	NSUN2-mediated methylation is associated with the mRNA export adaptor ALYREF’s nuclear-cytoplasmic shuttling, RNA-binding affinity, and associated mRNA export.	[36]
mRNA Stabilization	Methylation of the 3’-UTR stabilizes p16^INK4^/CDKN2A mRNA	[46]
	hypermethylated m5C mRNAs are recognized by YBX1 and then stabilized. by recruiting ELAVL1.	
tRNA Stabilization	Loss of Dnmt2 and NSun2 in mice results in substantially reduced steady-state tRNA levels.	[52]
tRNA Cleavage	m5C-methylation at the variable loop protects tRNAs from cleavage.	[41]
miRNA Processing and Cleavage	Methylation by NSUN2 affects processing and cleavage of pri- and pre-miRNAs.	[48]
Enhanced mRNA Translation	Methylation of various mRNAs is associated with enhanced translation.	[26,50]

**Table 2 cells-09-01758-t002:** Systematic analysis results for NSUN2 in lung cancer as determined using Lung Cancer Explorer [114].

**Meta-Analysis of Standardized Mean Difference of Tumor-Normal Gene Expression**											
Entrez ID	Symbol		SMD		SMD lower		SMD upper		pv	p.adj	Tumor-Normal Standardized Expression Difference
54888	NSUN2		1.65		1.44		1.85		2.5e-56	4.1e-54	Adenocarcinoma
			2		1.63		2.37		2.1e-26	8.2e-25	Squamous Cell Carcinoma
Entrez ID –National Center for Biotechnology Information (NCBI) designated Gene ID; Symbol—Gene Symbol; SMD—tumor-normal standardized mean difference; SMD.lower—lower bound of 95% confidence interval for SMD; SMD.upper—upper bound of 95% confidence interval for SMD; pv—p-value; p.adj—multiple comparison adjusted p-value by Benjamini Hochberg procedures											
**Meta-Analysis of Survival Association Based on Cox Proportional Hazards Model.**											
Entrez ID		Symbol		HR		Z		pv		p.adj	Tumor-Normal Standardized Expression Difference
54888		NSUN2		1.16		3.52		0.00043		0.0035	Adenocarcinoma
				1.07		1.4		0.16		0.76	Squamous Cell Carcinoma
Entrez ID -NCBI designated Gene ID; Symbol—Gene Symbol; HR—Hazard Ratio; Z—Z score from survival association meta-analysis; pv—p-value; p.adj—multiple comparison adjusted p-value by Benjamini Hochberg procedures											

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
