# Peer review of "The RNA Methyltransferase NSUN2 and Its Potential Roles in Cancer"

_cells, 2020, doi:10.3390/cells9081758_

Round 1

Reviewer 1 Report

The review manuscript by Chellamuthu and Gray entitled “The RNA methyltransferase NSUN2 and its potential roles in cancer” thoroughly describes the role of NSUN2 in several cancers. I do not have any issues with the content of the manuscript but have a significant number of issues with grammar, formatting and phrasing. These are thoroughly outlined below.

Capitalization issues

Line 44: decapitalize “Family” and “Methyltransferases”

Table 1 row 2: change “Mitotic Spindle Stability” to “Mitotic spindle stability”

Line 83: decapitalize “Disease”

Line 87: capitalize Drosophila

Line 119: decapitalize “Cancer”

Line 132: decapitalize “Breast” and “Cancer”

Line 157: decapitalize “Colorectal” and “Cancer”

Line 165: capitalize “circular”

Line 180: decapitalize “Lung and “Cancer”

Line 213: decapitalize “Analysis”

Lines 215 and 221: decapitalize all words except first

Table 2 last column: decapitalize all words except first

Figure 5: decapitalize “Lung” and italicize “n” referring to sample size

Figure 6: italicize and decapitalize “P” referring to p-values

Line 259: decapitalize “Other” and “Cancers”

Line 285: decapitalize all words except “Therapeutic” and “NSUN2”

Line 286: decapitalize “Kinases”

Table 1 row 5: change “PAR2 mediated Cell Migration” to “PAR2-mediated cell migration”

Table 1 column 1: be consistent in your capitalization of words, either capitalize every word or only capitalize the first word

Line 236: capitalize “supplementary” and “figure”

Awkward phrasing issues

Line 58-90: awkward phrasing, change to “roles for RNMTs in methylating mRNAs, miRNAs and lncRNAs have been predominately ascribed to NSUN2”

Line 105: does not make sense, revise to “overexpression of NSUN2 could rescue tau-induced toxicity”

Line 114: “methylation of ICAM-1 mRNA NSun2 promotes” does not make sense

Line 126-127: change “emerging data regarding NSUN2 in cancer” to “emerging data regarding the role of NSUN2 in cancer”

Line 158: does not make sense, do you mean “In addition to their original description of NSUN2 as an RNMT, Frye and Watt…”

Line 159: change “and subsequently confirmed” to “which was subsequently confirmed”

Line 169: remove the word “patients” that follows “(CRC)”

Line 173-176: does not make sense

Line 188-190: rephrase to “while some evidence has shown a role for NSUN3 in nuclear transcriptional regulation in leukemic cells”

Line 191: change “found constitutively” to “constitutively expressed”

Line 192: change “In knockout mice” to “In NSUN2 knockout mice”

Line 193: change “smaller than normal but show defects” to “smaller than normal and show defects”

Formatting issues

Abbreviations should not be defined in the abstract unless necessary. Please remove the following abbreviations: tRNAs, rRNAs, eRNAs, lncRNAs, mRNAs, RNMT, NSCLC. These should then be defined in the main text at their first use.

Italicize “et al.” throughout

Line 64-82 excl. table: one or two sentences cannot form a standalone paragraph. Consider collapsing all four of these small paragraphs into one big paragraph as they all discuss the various cellular functions of NSUN2. Same problem on lines 80-82, 84-85, 155-156, 225-226, 255-258 and 328-334.

Line 124: “TCGA” should be spelled out before abbreviating

Line 136: “eighteen” should be “18”

Line 137: “sixteen” should be “16”

Line 143: the “n” in parentheses should be italicized

Line 167: change “circular RNA (circRNA)” to “circRNA” as it’s already been defined above

Line 218, 219, 223: the “p” in “p-value” should be italicized

Line 241: remove “Clinical Proteomic Tumor Analysis Consortium (CPTAC)” and just use “CPTAC” because you already defined the acronym above

Line 305-306: change “cytosine-5 methylation” to “m5C”

Line 374: remove the phrase “Please add:”

Grammatical issues

Line 40: remove “the”

Lines 43, 69, 85, 274, 284: missing a “.”

Line 76 change “stress” to “stresses”

All instances of “NSUN2 mediated” should be changed to “NSUN2-mediated”

All instances of “NSUN2 directed” should be changed to “NSUN2-directed”

All instances of “PAR2 mediated” should be changed to “PAR2-mediated”

Table 1 row 4: change “translational” to “translation” and remove period

Table 1 row 9: change “steady state” to “steady-state”

Table 1 row 12: change “are” to “is”

Line 109: change “with” to “in”

Line 122: “on” should be changed to “upon”

Line 126: add a space between “Figure S1).” and “Over”

Line 143: remove dash in “up-regulated” and “up-regulation” to be consistent with your formatting in the rest of the manuscript

Line 146: remove dash in “hypo-methylation”

Line 150: add comma after “NSUN2”

Line 160: change “(PAR2) mediated” to “(PAR2)-mediated”

Line 162: remove dash in “down-regulation”

Line 170: add dash between “patient” and “derived”

Line 198: add comma between “samples” and “respectively”

Line 229: add space after period in “survival.All”

Line 245: add a space after period in “lung cancer.The”

Line 249: add “and” before “we” in “LUAD, we examined”

Line 277: change “Critically” to “Importantly”

Line 279: remove duplicate “in”

Line 290: change “Hela” to “HeLa cells”

Line 300: change “noncoding” to “non-coding” for consistency

Line 309: change “demonstrating from” to “demonstrated by” and italicize “in vitro”

Lines 315-319: run-on sentence, split into at least two sentences

Line 327: add a dash between “YBX1” and “mediated”

Author Response

Major changes for the editor/reviewers as per line number

Lines 42-98 – Reformatting at the request of Reviewer1 and additional text at the request of Reviewer 2

Lines 101-109 – Restructuring of section at the request of Reviewer 2

Lines 130-150 – Restructuring of paragraphs at the request of Reviewer 1

Lines 207-212 – Rephrasing of sentence to increase clarity.

Lines 258-268 – Additional text explaining how the analysis of the data in Table 2 was conducted at the request of Reviewer 2.

Lines 285-290 Additional text explaining the methodology used to generate Table 2 at the request of Reviewer 2.

Lines 471-492. Conclusion re-written to include the suggestions of Reviewer 2.

All other changes are simple typographical edits raised by Reviewers 1 & 2.

Reviewer 2 Report

1) General comments

The authors describe a very timely topic related to epitranscriptomics and its potential role for diseases. In addition to the current review I would like to see (1) a review and/or discussion of the molecular impact of m5C on the functionality of the RNA molecule (currently the functionality is exclusively linked to the RNA methyltransferase NSUN2), and (2) a review of how and how precise m5C modifications can currently be detected.

2) Specific comments for revision:
a) major
* in several occasions sentences should be rewritten because their meaning is unclear, e.g. lines 173-176, lines 336-337, and many more
b) minor
* section "2.1. Known cellular roles for NSUN2 directed RNA methylation": NSUN2's functionality should be clearly differentiated between direct function and indirect function through RNA methylation, e.g. "mitotic spindle stability" is a direct function of the protein whereas mRNA stability is a indirect function
* Table 1: some references are unaligned to the biological function
* line 105: typo
* conclusions: please rewrite with arguments why m5C modifications are of therapeutic interest against cancer and why specifically NSUN2 is of interest
* how was the data in table 2 conducted? The header of the last column in the table about Meta-analysis of Survival Association based on Cox Proportional Hazards Model seems to be wrong?
* resolution of Figure 4 is bad. no expanation of the figure in text.
* l.291: do not use the idiom "In a similar vein"
* I would suggest to add a paragraph how to actually look for m5C modifications
* caption is missing for supplementary figures. legend is missing for suppl. fig. 1

Author Response

(The authors gave the same response as above.)
